# Radiomics in Differentiated Thyroid Cancer and Nodules: Explorations, Application, and Limitations

**DOI:** 10.3390/cancers13102436

**Published:** 2021-05-18

**Authors:** Yuan Cao, Xiao Zhong, Wei Diao, Jingshi Mu, Yue Cheng, Zhiyun Jia

**Affiliations:** 1Department of Nuclear Medicine, West China Hospital of Sichuan University, Chengdu 610040, China; 2019224020064@stu.scu.edu.cn (Y.C.); 2018324025240@stu.scu.edu.cn (X.Z.); diaowei@stu.scu.edu.cn (W.D.); vivienne0117m@163.com (J.M.); 2Department of Radiology, West China Hospital of Sichuan University, Chengdu 610040, China; 2019224025311@stu.scu.edu.cn

**Keywords:** differentiated thyroid cancer, radiomics, ultrasound, magnetic resonance imaging, computer tomography, prediction, classification

## Abstract

**Simple Summary:**

Differentiated thyroid cancer (DTC) is the most common endocrine malignancy with a high incidence rate in females. The COVID-19 epidemic posed an increased risk of treatment delay causing increased DTC morbidity and mortality rate of DTC. Several imaging techniques, including ultrasound (US), magnetic resonance imaging (MRI), and computer tomography (CT), have been applied in the early screening and diagnosis of DTC. However, these traditional methods have limited sensitivity and specificity due to dependence on the experience and skill of the radiologists.

**Abstract:**

Radiomics is an emerging technique that allows the quantitative extraction of high-throughput features from single or multiple medical images, which cannot be observed directly with the naked eye, and then applies to machine learning approaches to construct classification or prediction models. This method makes it possible to evaluate tumor status and to differentiate malignant from benign tumors or nodules in a more objective manner. To date, the classification and prediction value of radiomics in DTC patients have been inconsistent. Herein, we summarize the available literature on the classification and prediction performance of radiomics-based DTC in various imaging techniques. More specifically, we reviewed the recent literature to discuss the capacity of radiomics to predict lymph node (LN) metastasis, distant metastasis, tumor extrathyroidal extension, disease-free survival, and B-Raf proto-oncogene serine/threonine kinase (BRAF) mutation and differentiate malignant from benign nodules. This review discusses the application and limitations of the radiomics process, and explores its ability to improve clinical decision-making with the hope of emphasizing its utility for DTC patients.

## 1. Introduction to Thyroid Cancer

### 1.1. The Epidemiology and Pathophysiology of Thyroid Cancer 

Thyroid cancer is the most common endocrine malignancy and the most commonly diagnosed cancer in people aged 15 to 29 years, and its incidence has continuously increased with 567,233 cases and approximately 41,000 deaths worldwide in 2018 [1,2]. The incidence rate of thyroid cancer is approximately three-fold higher in females than in males but the mortality rate is higher in males than in females [3,4,5]. In addition, a recent study confirmed that the recurrence rate of well-differentiated thyroid cancer (DTC) is higher in men compared with women. Due to the COVID-19 epidemic, delayed investigations and treatment may further lead to increased morbidity and mortality of thyroid cancer [6]. The various clinical outcomes of thyroid cancer are considered to be related to patient age, sex, tumor type, distant metastases, and invasion of adjacent tissue and lymph nodes [7]. 

Thyroid tumors are divided into follicular-derived and neuroendocrine C-cell-derived cancers. Greater than 95% of thyroid cancer is DTC, which is follicular-derived thyroid cancer and can be further divided into well-DTC and poorly-DTC (more progressive than well DTC) [8]. Well DTC is a composite of papillary thyroid cancer (PTC), follicular thyroid cancer (FTC), and Hurthle cell thyroid cancer. Of these, papillary thyroid cancer is the most common thyroid cancer with the best prognosis, whereas follicular, Hurthle cell, poorly-differentiated, and C-cell derived thyroid cancers are relatively uncommon but have a high metastatic risk to the lung and bone [7]. Moreover, the increasing diagnostic rate of papillary thyroid cancers is regarded as the leading reason for increasing thyroid cancer incidence, in contrast, the incidence rate of other subtypes has been stable in the past 30 years [9]. 

Despite the generally stable course, favorable prognosis, and low mortality of thyroid cancer, the rate of local recurrence and distant metastases of DTC approaches 10% to 30%, which depends on the length of follow-up [10,11]. A previous study found that DTC can recur even up to 20 years after the initial diagnosis [12], therefore, a long-term follow-up of patients with DTC is essential [13]. Notably, several studies have investigated the factors related to DTC relapse. However, heterogenicity exists among these studies, and the results indicated the associations between early-onset and recurrence [14]. More specifically, the earlier DTC occurred, the more likely it was to recur. Therefore, timely diagnosis of DTC recurrence is critical.

### 1.2. Imaging Techniques for DTC Detection

Additionally, the discrimination and identification of thyroid cancer nodules and thyroid benign nodules are important. In most cases, the initial presentation of thyroid cancer is a thyroid nodule [15], however, less than 10% of DTCs appear in the thyroid nodules [16]. Given various factors including age, sex, family history, exposure to radiation, and nodule size that could affect the shift from thyroid nodules to cancer [17], responding to this shift in a timely manner is necessary. Differentiating early malignant tumors from benign tumors and providing definite staging are key challenges for diagnosing and treating thyroid cancer. Moreover, estimating tumor progression or predicting prognosis precisely can significantly aid physicians in making clinical decisions regarding treatment strategies in patients with thyroid cancer. Palpation of the thyroid and cervical lymph nodes remains easiest and least expensive routine detection method, but this method is also the least sensitive [18,19]. In contrast, biopsy and histopathological examination are typically the diagnostic gold standard for thyroid cancer [20]. However, fine-needle aspiration biopsy (FNAB) usually samples a small portion of the lesions; thus, this method could provide limited information regarding tumor heterogeneity and may lead to missed diagnoses. Notably, a proportion of patients are still intractable to invasive examination for screening making it difficult to repeat pathological assessments. Noninvasive imaging examinations including high-resolution ultrasound, magnetic resonance imaging (MRI), computed tomography (CT), single-photon emission computed tomography (SPECT), positron emission tomography (PET), and PET/CT, are also playing an increasingly important role in initial tumor screening, staging, restaging, management, and posttreatment follow-up [21]. Of these, high-resolution ultrasound remains the sole fundamental imaging method in the diagnosis and screening of thyroid nodules and cancer [22]. High-resolution ultrasound is a safe noninvasive imaging technique that could aid in enhancing the early detection of pathologies [23]. 

Ultrasound is based on the pulse-echo principle that makes it possible to determine the thyroid size, location, number, and morphology of individual nodules, and occult nodules omitted by physical examination [18], and present these findings in a single cross-sectional B-scan image. The suspicious ultrasound features of malignant thyroid nodules exhibit the following characteristics: solid nodule structure, hypoechogenicity, taller-than-wide shape, irregular margin, microcalcification, and invasion of surrounding tissue [18,24]. Nodules that present pure cystic or cystic components that represent greater than 50% of nodule volume tend to be benign [25]. Despite its widespread availability and radiation-free features, the diagnosis significantly relies on the radiologists’ experience and subjective judgments, which limits the ability to make an objective diagnosis. Although neck ultrasound is the primary method used to investigate palpable thyroid masses, suspicious neck masses are typically initially screened through CT or MRI examination. Some features may be specific, whereas others may be incidental findings [26]. CT and MRI provide evidence for detecting lymph node metastasis as well as evaluating the invasion of adjacent tissue and organs with the features of cross-sectional imaging and reconstruction function [27]. It has been reported that the CT use rate in the examination of the neck and cervical spine has increased rapidly and is greater than that of ultrasound in the United States [28]. A previous study also verified the value of CT in detecting incidental thyroid nodules and hypothesized that CT imaging may be the current trend rather than ultrasound [29]. Nevertheless, CT also has an obvious limitation given that contrast-enhanced CT with iodinated contrast medium would delay subsequent radioactive iodine therapy [30]. Conversely, based on gadolinium-based contrast agents, MRI can be employed without interfering with radioiodine administration despite the requirement for a longer scan time. In addition, with its higher soft-tissue contrast, MRI combined with diffusion-weighted imaging (DWI) sequences could provide qualitative and quantitative information about tumor lesions at the cell level. DWI has been applied to assess the differentiation of benign and malignant thyroid tumors for several years [31,32]. A recent study verified the potential advantages of DWI in predicting aggressive histological features of thyroid carcinoma [33]. 

Nuclear medicine examinations have been used in the diagnosis, treatment, and surgical management of thyroid disease. Given the high avidity of radioiodine in functioning thyroid tissues, 131I whole-body scintigraphy (WBS) has a high value in tumor and metastasis detection [34]. WBS is also regarded as a routine diagnostic procedure for DTC patients with thyroidectomy [35]. However, WBS cannot provide a precise anatomic location, which greatly constrains its potential value. Compared with WBS, SPECT/CT not only enables anatomic localization of the tumor but also has higher sensitivity (50%) and specificity (100%) [36]. Besides, PET/CT or PET/MRI is also a relatively high sensitivity imaging technique in the evaluation of recurrent or metastatic tumors. A meta-analysis calculated that the pooled sensitivity was as high as 93.5% for PET/CT in detecting recurrent or metastatic DTC [37]. In addition, PET/CT can detect 21.2% of lymph nodes and soft tissue lesions that were missed by ultrasound [38]. Compared with PET/CT, PET/MRI has low radiation, but the high costs of this method should also be considered. Although some reports have demonstrated the value of the nuclear medicine approach applied in thyroid cancer, underlying issues, such as cost-effectiveness, universality, and radiation, require further discussion. Notwithstanding the above strengths, the current imaging technologies for tumor classification and prediction remain limited. Radiomics is an emerging field that involves segmenting lesions, extracting quantitative radiology features from medical images, and constructing models to classify or predict disease. The current review focuses on the radiomics characteristics of DTC and reviews the classification and prediction ability of radiomics for DTC.

## 2. Introduction to Radiomics

### 2.1. The Definition of Radiomics

Radiomics is defined as quantitative mapping that is used to construct a prediction model by extracting and analyzing medical image features related to the prediction target, including clinical endpoints and genomic features [39]. Radiomic features capture tissue and lesion characteristics, such as heterogeneity and shape, and may be used for clinical problem solving alone or in combination with demographic, histologic, genomic, or proteomic data. As an important innovation, medical image analysis automatically extracts a large number of quantitative features of medical images in a high-throughput manner. The use of radiomics in medical image analysis represents a significant improvement [40]. Radiomics research is based on the hypothesis that this type of automatic or semiautomatic software can provide better analysis of medical image data than human doctors due to the increased number of image features revealed by conventional and novel medical imaging that cannot be recognized by human doctors [41]. More specifically, the technology is based on the hypothesis that genomics and proteomics patterns can be expressed in terms of macroscopic image-based features [40].

### 2.2. Radiomic Features 

Compared with the so-called “semantic” qualitative features, which are typically subjectively defined by radiologists, radiologic features can be regarded as quantitative features and are generally divided into shape, first-order statistics, second-order statistics, and higher-order statistics [42]. Familiarity with core principles of radiomic features may facilitate interpretation of results and preselection of features for specific applications.

Shape features represent geometric relations that mainly refer to two-dimensional or three-dimensional image features derived from ROIs, such as tumor volume, surface area, tumor sphericity, and tumor compactness [43]. 

The first-order statistics features or histogram-based features are derived from the statistical moments of the image intensity histogram and based on the image intensity distribution represented by histograms that characterize the distribution of individual pixel or voxel intensity values within. Features, such as uniformity, asymmetry, kurtosis, and skewness, can also be used to extract other features, such as image energy and entropy [43,44]. 

Second-order statistical features, which are also known as texture features, quantify intratumoral heterogeneity and explain the spatial interdependence or cooccurrence of information between adjacent voxels [42]. Textural features are not directly computed from the original image but from different descriptive matrices that already encode specific spatial relations between pixels or voxels in the original image. In the original image, there are some matrices of the spatial relationship between the intensity of the encoded image from which a large number of texture features can be calculated. The gray value distribution matrix (GLCM) of cooccurrence voxels in the gray level co-occurrence matrix is one of the most commonly used second-order features in radiomics [45,46]. The neighborhood gray-level different matrix (NGLDM) and the gray-level run-length matrix (GLRLM) are also common. Higher-order statistical features are typically calculated using statistical methods after applying a specific mathematical transformation (filter). For example, repeating patterns, noise suppression, edge enhancement, histogram-oriented gradients, or local binary patterns (LBPs) can be identified. The applied mathematical transformations or filters include Laplacian transformations of Gaussian-filtered images (Laplacian-of-Gaussian), wavelet or Fourier transformations, Minkowski functionals, or fractal analysis [47].

### 2.3. The Workflow of Radiomics

Radiomics analysis can be achieved by two methods. The first method includes conventional and common typical methods that are used to determine the region of interest (ROI) of the medical image first and then extract the radiomics features from the ROI and analyze the clinical problem [48]. The second method is less applied but pointed out by the previous review, it works based on the radiomics images directly but not the radiomics data derived from conventional images, it is also helpful to recognize ROIs reliably [41].

Radiomics analyses begin with the choice of a disease and image protocol. When targeting disease and image protocols are selected, the classical radiomics process can be divided into the following four steps: selection of the regions of interest, radiomics feature extraction, analysis, and modeling [49]. Figure 1 illustrates the workflow of radiomics for thyroid disease.

ROIs are commonly delineated by professional radiologists manually or by special software in a semiautomatic or fully automatic manner. In the feature extraction stage, hundreds of candidate radiomic features are typically extracted theoretically to be used as the input of the prediction model, but the number of model parameters will increase exponentially afterward. Moreover, radiation features generally show a high degree of correlation with each other, indicating data redundancy. Thus, some features can be discarded, whereas other features can be grouped and replaced by representative features. Therefore, a large number of candidate features must be removed or transformed via a process called dimensionality reduction [47].

After feature selection, a mathematical model can be established to predict or solve targeted medical problems, such as the existence of specific gene mutations or tumor recurrence. Radiation features can be modeled in many different ways, ranging from statistical models to machine learning methods, depending on the clinical problems to be solved [50]. The most popular algorithms in radiomics are linear regression and logistic regression, decision trees (such as random forests), support vector machines (SVMs), neural networks, and Cox proportional hazards models with censored survival data.

### 2.4. Clinical Applications of Radiomics

The application and research potential of radiomics are still being explored. However, based on published studies, the clinical application of radiomics can be classified into the following three categories: radiogenomics (linking imaging data to biology), diagnosis of diseases, and clinical outcome prediction, including treatment response, recurrent disease, and survival time [47,51]. However, radiomic studies of thyroid cancer mainly involve the latter two categories.

## 3. Literature Search Strategy

We conducted a comprehensive literature review from the PubMed, Web of Science and Google Scholar databases for papers published before February 2021, independently. English-language filters were applied in the process of searching. Standard searches were done with the following keywords: ‘thyroid cancer’, ‘differentiated thyroid cancer’, ‘thyroid nodules’, and ‘radiomics’. The reference lists were manually checked to identify additional relevant studies. We followed The Preferred Reporting Items for Systematic Reviews and Meta-Analysis (PRISMA) guidelines to select relevant studies [52] (Figure 2).

## 4. Radiomics in Thyroid Cancer Prediction

As mentioned above, radiomics aids in cancer detection, diagnosis, prediction of prognosis, evaluation of tumor status, treatment response, and local or distant metastasis [50]. Of these, the predictive value has been determined in various cancers and has been a research hotspot in recent years. Table 1 showed the predictive value of radiomics applied in DTC, Table 1 was organized according to a sequential order of prediction category, imaging method, and published time.

Metastasis is an important indicator of tumor progression [53]. Lymph node (LN) metastasis is closely related to local recurrence, distant metastasis, and thyroid stage, which further indicates the surgical plan [54,55]. Thus, the judgment of LN metastasis is important. Although a small proportion of patients report LN metastasis, those patients with suspicious abnormalities would also be suggested to undergo fine-needle aspiration biopsy (FNA) and prophylactic lymph node dissection (LND). These invasive examinations seem to be unsuitable for those people without LN metastasis. Therefore, it is important to identify a noninvasive approach to pinpoint patients with high-risk LN metastasis in clinical practice. Liu et al. [56] compared the radiomics prediction ability to estimate the LN status among B-mode ultrasound (B-US), strain elastography ultrasound (SE-US) images, and the combination of these two images. As was hypothesized, the combination group showed a better prediction ability than a single image. However, given that only 75 patients were recruited and no validation analysis was performed in this study, the results should be interpreted with caution. Furthermore, the same research team included 450 patients and divided them into training and validation datasets to verify the radiomics evaluation of US thyroid images to predict LN metastasis in PTC patients [57]. This study partly validated their previous conclusion that the features ultimately selected performed equally well regarding the radiomics evaluation. PTC patients with or without LN metastasis showed different radiomics signatures. Jiang et al. [58] extracted radiomics features from both shear-wave elastography (SWE) images and B-mode ultrasound (BMUS) images. They calculated the Rad-score to distinguish patients with high metastasis risk. Then they built and compared the value of radiomics nomogram and clinical nomogram in predicting the LN stage. They concluded that the nomogram based on SEW radiomics signatures performed well in predicting LN status. Li et al. [59] also verified the value of ultrasound radiomics features in predicting LN metastasis. The radiomics features had a larger AUC than the ultrasound features of microcalcifications and an irregular shape. 

Although CT and MRI are not exceedingly superior to ultrasound in thyroid cancer diagnosis, CT-based and MRI-based radiomics performed equally as well regarding their predictive value. The ability of CT radiomics signature to predict LN metastasis was initially reported by Lu et al [60]. This group built an SVM model and found that the radiomics signature showed a better predictive value of LN metastasis than any single radiomics signature. They concluded that the radiomics nomogram adds predictive power to LN metastasis. Hu et al. [61] initially applied multimodal MRI radiomics to predict LN metastasis in patients with PTC, and Zhang et al. [62] extracted radiomics features from T2WI and T2WI-fat-suppression (T2WI-FS) images to test and validate the predictive value of LN metastasis. These studies partly demonstrated that MRI-based radiomics can scientifically, quantitatively, and accurately predict LN metastasis in PTC patients, thereby, reducing unnecessary surgery. 

LN metastasis is more likely to occur in central regions followed by lateral regions [3]. Lateral LN metastasis exhibits a higher recurrence rate and a poorer prognosis than central LN metastasis [63,64]. A recent study developed an ultrasound-based radiomics nomogram to assess its predictive value for central neck lymph node metastasis in PTC patients [65]. The prediction model showed good accuracy, sensitivity, specificity, and AUC values in both the training dataset and validation dataset. Afterward, the predictive value of ultrasound radiomics for lateral cervical LN metastasis was successively investigated in two studies. Tong et al. [66] retrospectively recruited 840 patients with PTC and extracted radiomics features from their preoperative ultrasound images. These researchers also established a radiomics-based nomogram to predict lateral LN metastasis. This radiomic nomogram presented good discrimination in both training and validation datasets and may therefore have clinical application. More interestingly, one study found a link between ultrasound radiomic features of the primary tumor and the status of lateral LN metastasis [67]. The key and interesting part of this study was that it focused on the radiomics features of thyroid primary tumors in predicting lateral LN metastasis but not the LN itself, which may facilitate the early detection of metastases. 

Although the results of the abovementioned studies on the predictive value of ultrasound radiomics were largely positive in nature, the main limitation of the lack of multicenter and external validation could not be overlooked. A recent relatively robust study filled this gap. Yu et al. [3] first focused on the diagnostic value of ultrasound radiomics under a multicenter, cross-machine, multi-operator scenario. Based on B-mode ultrasound images of thyroid lesions, they established and compared four models including clinical statistical model (SM), traditional radiomics model (RM), non-transfer learning model, and transfer learning radiomics (TLR) model to predict the risk of LN metastasis in PTC patients. Of these, the TLR model showed the highest sensitivity and specificity in both the main and external cohorts. Then, a recent study that is in preprint performed an external validation based on CT radiomics indicating the good performance of this method in the prediction of LN metastasis [68]. To some extent, this study adds strength and validity to previous ultrasound-based radiomics studies. 

Besides, the predictive value of radiomics was also applied in other aspects, such as the prediction of distant metastasis [69], tumor extrathyroidal extension [70,71], disease-free survival [72], and BRAF mutation [73]. The aggressiveness of tumors is classified based on various features, such as extrathyroidal extension; aggressive pathological subtypes, such as tumors with tall cells, tumors with columnar cells, and the hobnail variant; lymph node involvement; and distant metastasis [74]. A recent study found that multiparametric MRI-based radiomics combined with a machine learning approach can accurately distinguish aggressive PTC patients from nonaggressive patients, which illustrated the role of radiomics in predicting aggressive tumors [75]. Distant metastasis of DTC is uncommon; however, FTC is more likely to have distant metastasis than PTC. It has been reported that the bone metastasis rate in FTC ranges from 7 to 28%, whereas that for PTC is only 1.4–7% [76]. Kwon et al. [69] thus evaluated the capability of ultrasound-based radiomic features to predict distant metastasis of FTC. This study is based on radiomics analysis and a machine learning approach, and multivariate analysis indicated that the radiomic signature and widely invasive histology are related to distant metastasis. Moreover, the AUC of the thyroid ultrasound radiomic signature in predicting distant metastasis was as high as 0.93, demonstrating good predictive performance. The extrathyroidal extension in patients with DTC is also an important factor to consider when determining the surgical modality. Chen et al. [70] selected five CT-based radiomics features that were closely related to the extrathyroidal extension of PTC patients. A CT-based radiomics nomogram was built and showed good predictive value in extrathyroidal extension. This excellent predictive performance for tumor extrathyroidal extension was also verified in an MRI-based radiomics preprint [71]. Regarding “disease-free” cancers, DTC has an overall good disease-free survival after treatment and long-term outcomes [77]. Despite being called a “happy cancer”, tumor progression contributes to the 1.4–5.2% mortality rate of thyroid cancer [78,79]. A retrospective study included 768 PTC patients, extracted radiomics features from ultrasound images, and constructed a radiomics signature based on LASSO regression. Finally, a Rad-score was calculated to stratify the patients into high- and low-risk DFS [72]. Furthermore, based on recent progress in molecular genetics, gene-specific information has provided insights into the biology of the tumor, prediction of prognosis, and potential therapeutic targets [80]. The B-Raf proto-oncogene serine/threonine kinase (BRAF) mutation is involved in the pathogenesis of PTC and is related to tumor progression, recurrence, and mortality [73]. In addition, shedding light on the mutational status of thyroid cancer could help clinicians evaluate the tumor response to new drugs, such as tyrosine kinase inhibitors. Thus, if we can predict genes mutated in thyroid cancer through convenient and feasible approaches, this information would contribute to improving tumor diagnosis, judging the prognosis, and personalizing the treatment. To date, two studies have applied radiomics to estimate BRAF mutations in PTC patients [73,81]. These two studies offered a consistent outcome that ultrasound radiomics has a limited value in predicting BRAF nutation. This result indicated that the relationship between ultrasound radiomics and gene mutation may not be as good as expected.

## 5. Radiomics in Thyroid Cancer and Nodule Classification

Thyroid cancer nodules are common in thyroid disease. The prevalence of thyroid nodules is approximately 67% in adults [82]. It has been reported that approximately 10% of patients with detected thyroid nodules are diagnosed with malignancy [83]. LN metastasis indicates rapid progression and poor prognosis in thyroid cancer; however, only a small portion of patients will develop metastasis [56]. Therefore, special attention needs to be paid to the differentiation of malignant and benign nodules. Table 2 presents studies focusing on the classification value of radiomics in DTC.

Machine learning or deep learning-based modalities are an important step in the processing of radiomics data. Machine learning approaches typically manually select and delimitate a set of few ROIs on the images; then, machine learning algorithms, such as support vector machine (SVM), random forest, and least absolute shrinkage and selection operator (LASSO), are applied to build a model. Prochazka et al. [84] used histogram analysis and segmentation-based fractal texture analysis algorithms combined with SVM and random forest classifiers to distinguish malignant nodules from benign nodules in ultrasound images. Their results indicated that the histogram feature was the most important parameter in classification, and both SVM (94.64%) and random forests (92.42%) achieved high accuracy. Colakoglu et al. [85] attempted to differentiate benign and malignant thyroid nodules using texture analysis and random forest model construction. After testing the reproducibility of all texture features, they finally screened seven texture features from ultrasound images, including one histogram (HistPerc 99), one HOG (HogO8b2), four GRLMs (GrlmHRLNonUni, GrlmHMGLevNonUni, GrlmNRLNonUni, and GrlmZRLNonUni), and one GLCM (GlcmZ3AngScMom), in a random forest model. The diagnostic sensitivity, specificity, and accuracy were 85.2%, 87.9%, and 86.8%, respectively. Notably, the area under the curve (AUC) of the model was 0.92, indicating good performance. Furthermore, a recent ultrasound-image-based retrospective study recruited 2558 patients (2831 nodules), extracted radiomics features using an in-house texture analysis algorithm, and applied the LASSO method to calculate the radiomics score [86]. They used this radiomics score to determine a cutoff value that can help classify the nodules as benign or malignant. The AUCs of the radiomics score in the training and testing datasets were 0.85 and 0.83, respectively, indicating discriminative power. Furthermore, Yoon et al. [87] also applied texture analysis and the LASSO method in US images to predict malignant thyroid nodules with indeterminate cytology, demonstrating good predictive performance. Zhao et al. [88] compared the diagnostic performance and unnecessary FNAB rate for thyroid nodules of assisted visual-based and radiomic-based machine learning approaches in ultrasound images. In this study, ten machine learning classifiers, including decision tree, naïve Bayes, k nearest neighbors (KNN), logistics regression, SVM, KNN-based bagging, random forest, extremely randomized trees (XGBoost), multilayer perception, and gradient boosting tree classifiers, were verified. The results of the assisted visual-based machine learning approach indicated superior performance in AUC, sensitivity, and specificity in both the training dataset and internal validation dataset. Furthermore, a similar study design was applied to a CT-based radiomics study [89]. This study ultimately included 13 radiomics features after LASSO logistic regression. An SVM model was constructed and compared with seven other machine learning models. The study concluded that the SVM model exhibited good discrimination performance, whereas random forest had the highest stability.

In addition, the ability to discriminate benign from malignant lesions using a deep learning radiomics approach has also been verified by researchers. Zhou et al. [90] employed the deep learning radiomics method to differentiate benign and malignant thyroid nodules in ultrasound images. This study found that the AUC of deep learning radiomics was greater than that of other deep learning models and traditional naked-eye observations. However, the current limitation of deep learning is its black box issue, making the conclusion difficult to interpret. The abovementioned radiomic-based and deep learning-based classifications are two methods applied in the detection of malignant nodules and metastatic cervical lymph nodes. However, research comparing the diagnostic ability between these two approaches is insufficient but essential. Wang et al. [91] extracted 302-dimensional statistical features from ultrasound images and applied mutual information and linear discriminant analysis to reduce dimensionality. These researchers reported that the accuracy of radiomics for the testing data was 66.81%, which was relatively lower than that of the deep learning approach (74.69%). Although the radiomics approach may not be dominant in this study, the interpretability of deep learning is a long-standing problem that remains elusive.

## 6. Limitations

Substantial radiomics studies have indicated the predictive value of radiomics in DTC; however, it is undeniable that there are also several limitations in radiomics. First, the ‘black box’ property of classifiers hampers the causal relationship, and the meaning of radiomics features extracted from grayscale images further hinders data interpretability. Second, radiomics is regarded as a ‘population imaging’ approach closely relying on different modalities and device parameters, which means variations in imaging protocols among institutions would lead to non-uniform data acquisition and thus influence generalizability. The good classification and prediction performance in a single center might not be generalized to patient cohorts from another center. Therefore, current original studies generally lack external validation. Third, although radiomics partially reflects the information at the molecular biological level, variations in tumor cells and the microenvironment as well as the retrospective nature of the studies represent limit the interpretation of the final results. Notably, based on current studies, the average diagnostic accuracy of radiomics is between 66% and 86%, even worse in the prediction of BRAF mutation, making the economic efficiency is an issue in need of attention and consideration. Furthermore, the reliability of the predictive performance and clinical application may be decreased by discussing the predictive value of radiomics itself without considering the influence of clinical information, such as tumor stages and therapy strategies. More importantly, the ethical issues regarding the use of radiomics in patient stratification and treatment response-based prognosis should also be treated with caution.

## 7. Conclusions

In summary, radiomics is a hot topic and a rapidly evolving field in medical imaging in general. There are still some technological and ethical limitations of radiomics aforementioned are required to be solved. Nevertheless, increased studies have proved the potential applications of radiomics for both the research and clinical lactation field. For prediction, the radiomics is seemly satisfactory to predict lymph node metastasis, distance metastasis, tumor aggressiveness and extrathyroidal extension, and disease-free survival. While previous original studies consistently negated the value of US radiomics in predicting BRAF mutations in DTC. This result may need to be thoroughly discussed to determine the predictive value of other imaging techniques, such as CT and MRI, and provide a direct or indirect relationship between radiomics and tumor mutations of thyroid cancer in the future study. For diagnosis, the current findings may facilitate breakthroughs in thyroid cancer and nodule classification based on a radiomics approach. These studies demonstrated the usefulness of radiomics in discriminating benign and malignant lesions regardless of the image types (US, CT, or MRI). Further studies should address two important issues: (1) optimize the algorithm and models to improve the accuracy of external validation thereby enhance the diagnostic capacity of radiomics; (2) analyze multi-model or multi-parameters imaging data with a larger sample as well as increase the possibility of clinical transformation.

## Figures and Tables

**Figure 1 cancers-13-02436-f001:**
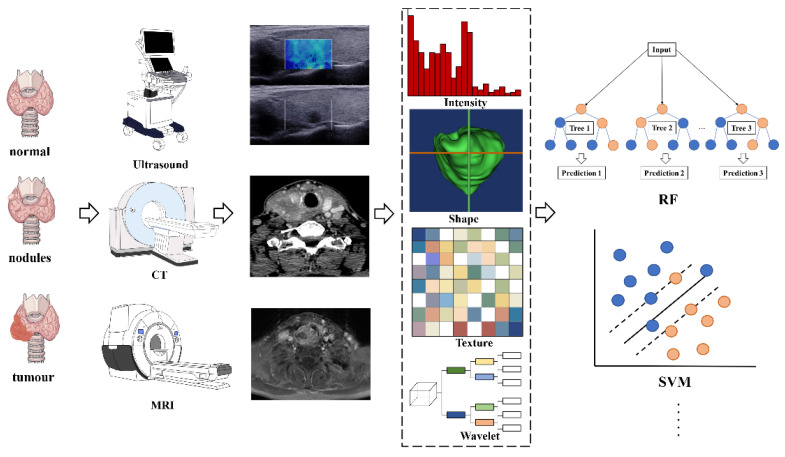
The flowchart shows the workflow of radiomics and its application in thyroid cancer or nodule classification and prediction. Abbreviations: RF—random forest; SVM—support vector machine

**Figure 2 cancers-13-02436-f002:**
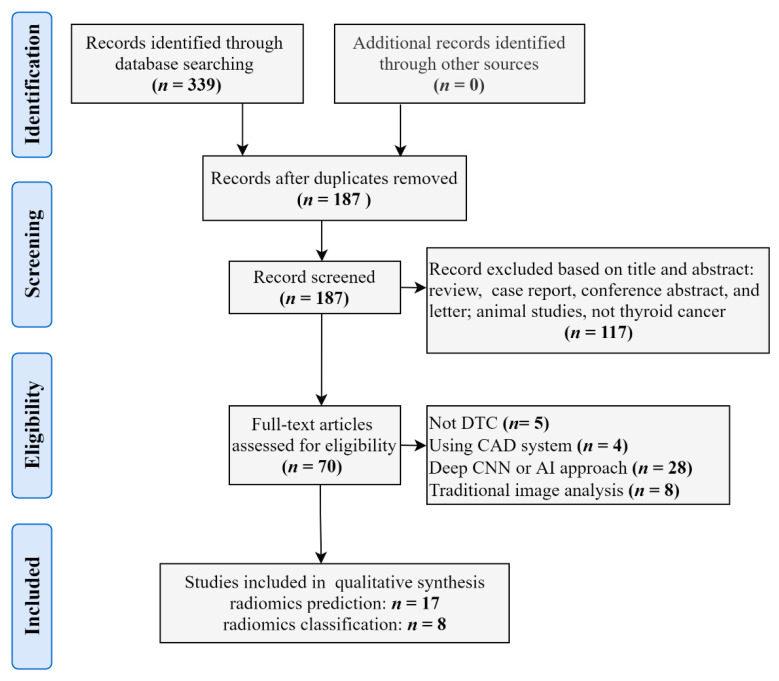
Flow diagram for the identification and exclusion of studies in radiomics application in differentiated thyroid cancer and nodules. Abbreviations: DTC—differentiated thyroid cancer; CAD—computer-aided detection; CNN—convolutional neural networks; AI—artificial intelligence.

**Table 1 cancers-13-02436-t001:** Studies used radiomics for the prediction of metastasis, tumor progression, treatment response, and gene mutation.

Reference	Prediction Category	No. Patients	Imaging Method	ROI Segmentation Method	No. Radiomics Features	Model Construction	Validation Method	Sensitivity (%)	Specificity (%)	Accuracy (%)	AUC
Liu et al. (2018) [56]	LNM	75	US and SEUS	manual	US + SWE: 25 US: 36 SWE: 9	SVM	LOOCV	US + SEUS: 77 US: 63 SEUS: 71	US + SEUS: 88 US: 89 SEUS: 75	US + SEUS:85 US: 83 SEUS: 74	US+SEUS: 0.90 US: 0.81 SEUS: 0.80
Liu et al. (2019) [57]	LNM	450	US images	manual	50	SVM	10-fold CV	67.9	72.5	71.1	0.783
Jiang et al. (2019) [58]	LNM	training: 147 EV: 90	SWE images	manual	4	LASSO logistic regression	10-fold CV	training: 80.67 EV: 86.84	training: 82.7 EV: 73.08	training: 78.91 EV: 78.89	training: 0.851 EV: 0.832
Li et al. (2020) [59]	LNM	126	US images	manual	91	hypothesis-testing and bagging	NA	training: 90 test: 72.7	training: 86 test: 80	NA	training: 0.759 test: 0.803
Zhou et al. (2020) [65]	LNM	training: 609 test: 326	US images	manual	23	LASSO logistic regression	NA	training: 82.5 test: 81.6	training: 78.6 test: 81.0	training: 79.8 test: 81.2	training: 0.87 test: 0.858
Tong et al. (2020) [66]	LNM	training: 600 test: 286	US images	manual	21	LASSO logistic regression	NA	training: 74.5 test: 77.4	training: 82.6 test: 83.1	NA	training: 0.877 test:0.862
Park et al. (2020) [67]	LNM	training: 400 test: 368	US images	manual	14	LASSO logistic regression	10-fold CV	NA	NA	NA	training: 0.71 test: 0.621
Yu et al. (2020) [3]	LNM	training: 1013 IT1: 368 IT2: 513	US images	manual	NA	TLR;SM;RM;NTLR	NA	SM: 72 (training); 43 (IT1); 68 (IT2) RM: 71 (training); 36 (IT1); 47 (IT2) NTLR: 75 (training); 71 (IT1); 67 (IT2) TLR: 94 (training); 83 (IT1); 95 (IT2)	SM: 82 (training); 87 (IT1); 67 (IT2) RM: 57 (training); 72 (IT1); 69 (IT2) NTLR: 81 (training); 81 (IT1); 78 (IT2) TLR: 77 (training); 89 (IT1); 75 (IT2)	SM: 77 (training); 61 (IT1); 67 (IT2) RM: 62 (training); 51 (IT1); 60 (IT2) NTLR: 79 (training); 75 (IT1); 73 (IT2) TLR: 84 (training); 86 (IT1); 84 (IT2)	SM: 0.83(training); 0.67(IT1); 0.67(IT2) RM: 0.64(training); 0.55(IT1); 0.57(IT2) NTLR: 0.82(training); 0.81(IT1); 0.79(IT2) TLR: 0.93(training); 0.93(IT1); 0.93(IT2)
Lu et al. (2019) [60]	LNM	training: 154 test: 67	CT	manual	8 radiomic sub-signatures	SVM	NA	NA	NA	training: 73.4 test: 64.2	training: 0.759 test: 0.706
Hu et al. (2020) [61]	LNM	training: 90 test: 39	MRI	manual	30	LASSO logistic regression	NA	T2WI model: 62.2 DWI model: 86.7 T1C+ model: 68.9 Combined model: 88.9	T2WI model: 87.2 DWI model: 70.2 T1C+ model: 83 Combined model: 72.3	T2WI model: 75.0 DWI model: 78.3 T1C+ model: 76.1 Combined model: 80.4	T2WI model: 0.819 DWI model: 0.826 T1C+ model: 0.808 Combined model: 0.835
Zhang et al. (2020) [62]	LNM	61	MRI	manual	10	RF	LOOCV	T2WI: 83 T2WI-FS: 83	T2WI: 100 T2WI-FS: 90	T2WI: 87 T2WI-FS: 82	T2WI: 0.85 T2WI-FS: 0.80
Kwon et al. (2020) [69]	DM	169	US images	manual	6	SVM	5-fold CV	training: 92 test: 80	training: 87 test: 87	training: 88 test: 85	training: 0.93 test: 0.90
Wang et al. (2019) [75]	Aggressiveness	120	MRI	manual	5	LSSO + GBC LSVM + LRCV LSVM + PAC LSVM + LSVC	10-fold CV	NA	NA	NA	train: 0.874; 0.979;0.971; 0.805; 0.974 test: 0.915; 0.731; 0.731; 0.885; 0.708
Chen et al. (2020) [70]	ETE	training: 437 test: 187	CT	manual	5	LASSO logistic regression	10-fold CV	NA	NA	NA	training: 0.791 test: 0.772
Park et al. (2019) [72]	DFS	768	US images	manual	40	LASSO COX regression	10-fold CV	NA	NA	NA	0.777 (C index)
Yoon et al. (2020) [73]	BRAF Mutation	training: 387 test: 140	US images	manual	8	LASSO logistic regression	NA	NA	NA	NA	training: 0.718 (C index) test: 0.629 (C index)
Kwon et al. (2020) [81]	BRAF Mutation	96 patients	US images	manual	43	logistic regression SVM RF	5-fold CV	66.8 (mRMR)	61.8 (mRMR)	64.3 (mRMR)	0.65 (mRMR)

Abbreviations: ROI—region of interest; AUC—area under the curve of receiver operating characteristic curve; LMN—lymph node metastasis; US—ultrasound; SEUS—strain elastography ultrasound; SWE—shear-wave elastography; CT—computer tomography; SVM—support vector machine; RF—random forest; LASSO—least absolute shrinkage and selection operator; CV—cross-validation; LOOCV—leave-one-out CV; EV—external validation; TLR—transfer learning radiomics; SM—statistical model; RM—traditional radiomics model; NTLR—non-transfer learning radiomics; IT—independent set; DM—distance metastasis; MRI—magnetic resonance imaging; ETE—extrathyroidal extension; DFS—disease-free survival; LSVM—linear support vector machine; LR—CV-logistic regression classifier with cross-validation; PAC—passive aggressive classifier; LSVC—linear support vector classification; mRMR—minimum redundancy maximum relevance; NA—not applicable.

**Table 2 cancers-13-02436-t002:** Studies used radiomics to differentiate malignant from benign nodules.

Reference	No. Patients/Nodules	Imaging Method	ROI Segmentation	No. Radiomics Features	Model Construction	Validation Method	Sensitivity (%)	Specificity (%)	Accuracy (%)	AUC
Prochazka et al. (2019) [84]	40 nodules in 40 patients	US images	threshold	NA	SVM/RF	LOOCV	NA	NA	NA	RF: 0.9242 SVM: 0.9464
Colakoglu et al. (2019) [85]	235 nodules in 198 patients	US images	manual	7	RF	10-fold CV	85.2	87.9	86.8	0.92
Park et al. (2020) [86]	1624 nodules in 1609 patients training: 1299; test: 325	US images	manual	66	LASSO logistic regression	10-fold CV	70.6	79.8	77.8	0.75
Zhao et al. (2020) [88]	training: 743 nodules in 720 patients test: 106 nodules in 102 patients	US and SWE images	manual	26	SVM	NA	74.4 (US) 70.7 (US + SWE)	72.3 (US) 79.4 (US + SWE)	73.1 (US) 76.2 (US + SWE)	US: 0.798 US + SWE: 0.834
Zhou et al. (2020) [90]	1750 nodules in 1734 patients	US images	semi-automated	NA	Deep learning	NA	training: 90.1 IV: 89.3 EV: 89.5	training: 82.7 IV: 83.5 EV: 84.1	NA	training: 0.96 IV: 0.95 EV: 0.97
Wang et al. (2020) [91]	3120 nodules in 1040 patients	US images	semi-automated /manual	302	SVM	NA	51.19	75.77	66.81	0.6371
Yoon et al. (2020) [87]	155 nodules in 154 patients	US images	manual	15	LASSO logistic regression	10-fold CV	NA	NA	NA	US + Clinical information: 0.839
Yao et al. (2020) [89]	1372 patients	CT images	manual	13	LASSO +RF	10-fold CV	68	82	74	0.82

**Abbreviations:** ROI—region of interest; AUC—area under the curve of receiver operating characteristic curve; SWE—shear-wave elastography; US—ultrasound; CT—computer tomography; SVM—support vector machine; RF—random forest; LASSO—least absolute shrinkage and selection operator; CV—cross-validation; LOOCV—leave-one-out CV; IV—internal validation; EV—external validation; NA—not applicable.

## Data Availability

No new data were created or analyzed in this study. Data sharing is not applicable to this article.

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
