# Peer review of "Radiomics in Differentiated Thyroid Cancer and Nodules: Explorations, Application, and Limitations"

_cancers, 2021, doi:10.3390/cancers13102436_

Round 1

Reviewer 1 Report

The authors are nicely reviewing the significance of radiomics in thyroid cancer, a new but promising field that may be proved useful not only for preventing useless extensive surgical resections but also treating eventually aggressive neoplasms. The only query for the authors is referring to the research strategy followed by the authors in order to select the specific studies to be presented. 

Author Response

Response 1: Thanks for your positive comments. We have added the research strategy and flow chart. Please see line 232, section 3 Literature search strategy highlighted in red

Reviewer 2 Report

General comments:

The article reviews a timely topic – of the use of radiomics in thyroid cancer for classification and prediction. The paper is generally well-written and incorporates a good number of studies to be able to evaluate the potential role of radiomics in the management of DTC.

The following are my suggestions to further improve the manuscript:

The paper would benefit from a better structuring. The introduction to thyroid cancer should be divided into two sections (1) anatomo-pathology (or similar) and (2) imaging techniques for DTC detection.

The studies presented in table 1 should be listed according to a system – either chronological order or based on the type of imaging used for radiomics, or other criteria chosen by the authors. This should also be mentioned in the text.

The limitations of radiomics should be discussed under a separate heading. The authors should also mention the challenges around variations in imaging protocols among institutions leading to non-uniform data acquisition; standardization / data harmonization; and ethical issues regarding the use of AI in patient stratification and treatment response-based prognosis.

Specific comments:

  1. Simple summary: the sensitivity and specificity of various imaging techniques are not operator-dependent but method dependent. The operator/radiologist brings a subjective factor due to differences in skills / experience.
  2. Line 34 - Introduction to thyroid cancer
  3. Line 44 – replace ‘tumor typing’ with ‘tumor type’
  4. Line 45 – remove ‘or not’
  5. Line 69 – replace ‘radiation explosion’ with ‘exposure to radiation’
  6. Line 76 – remove ‘easiest’ – it’s written twice
  7. Line 142 – Introduction to radiomics
  8. Section 2.3 – first paragraph – what is the second method by which radiomics analysis is achieved? You only mention that the second method is less applied, without expanding on it.
  9. Line 246 – As was hypothesized, the combination….
  10. Line 263 - … performed equally well regarding…

Author Response

Response to Reviewer 2 Comments

Point 1: General comments:

The article reviews a timely topic – of the use of radiomics in thyroid cancer for classification and prediction. The paper is generally well-written and incorporates a good number of studies to be able to evaluate the potential role of radiomics in the management of DTC.

Response 1: Thank you for your kind comments.

The following are my suggestions to further improve the manuscript:

Point 2: The paper would benefit from a better structuring. The introduction to thyroid cancer should be divided into two sections (1) anatomo-pathology (or similar) and (2) imaging techniques for DTC detection.

Response 2: We have revised our manuscript as you suggested (section of introduction in red)

Point 3: The studies presented in table 1 should be listed according to a system – either chronological order or based on the type of imaging used for radiomics, or other criteria chosen by the authors. This should also be mentioned in the text.

Response 3: We explained the basis for our arrangement of tables (line 251-253).

Point 4: The limitations of radiomics should be discussed under a separate heading. The authors should also mention the challenges around variations in imaging protocols among institutions leading to non-uniform data acquisition; standardization / data harmonization; and ethical issues regarding the use of AI in patient stratification and treatment response-based prognosis.

Response 4: We have discussed the limitation as a separate heading and mentioned the advice you suggested. Thank you for your deep insight into our manuscript, the suggestions you pointed out are important.  

Point 5: Specific comments:

Simple summary: the sensitivity and specificity of various imaging techniques are not operator-dependent but method dependent. The operator/radiologist brings a subjective factor due to differences in skills / experience.

Line 34 - Introduction to thyroid cancer

Line 44 – replace ‘tumor typing’ with ‘tumor type’

Line 45 – remove ‘or not’

Line 69 – replace ‘radiation explosion’ with ‘exposure to radiation’

Line 76 – remove ‘easiest’ – it’s written twice

Line 142 – Introduction to radiomics

Section 2.3 – first paragraph – what is the second method by which radiomics analysis is achieved? You only mention that the second method is less applied, without expanding on it.

Line 246 – As was hypothesized, the combination….

Line 263 - … performed equally well regarding…

Response 5: Thank you for your patience. We have modified each of the errors you pointed and highlighted them in red.

Reviewer 3 Report

Very well-written for most of the sections. Readers could really enjoy the flow of science. Just to add--please comment on the conclusion section. What will be the future and your own inputs on it. Please check on the tenses--past and past-perfect tense are commonly interchanged.

Author Response

Response to Reviewer 3 Comments

Point 1: Very well-written for most of the sections. Readers could really enjoy the flow of science. Just to add--please comment on the conclusion section. What will be the future and your own inputs on it. Please check on the tenses--past and past-perfect tense are commonly interchanged.

Response 1: Thanks for your positive comment. We have re-organized the section of conclusion as you and reviewer 4 suggested.

Reviewer 4 Report

The authors have made a comprehensive review about radiomics and differentiated thyorid cancer.

The aim of the review is noble however the structure of the final paper and especially the conclusion drawn by the authors are in my view disapponting for the following reasons:

1) The authors try to provide a very "all'-in-one" review about what is known for DTC but they spend almost half of the paper introducin generic aspects of radiomics not directly linked to DTC: in my view this is unnecessary

2) Reading the title of the paper the reader would expect to get information about the diagnosis of DTC but then the authors state they want to provide "insight into the capacity of radiomics to 24 predict lymph node (LN) metastasis, distant metastasis, tumor extrathyroidal extension, disease-25 free survival, and B-Raf proto-oncogene serine/threonine kinase (BRAF) mutation": in my view this is misleading

3) The conclusion is totally absent becauese atuhors state that the "aforementioned findings"demonstrate the usefulness of radiomics...regardless of the image type": such sentence is lacking scientific evidence beacuse US is the most widely used technique and should receive in my view a special focus both in the discussion and in the conclusion.

Author Response

Response to Reviewer 4 Comments

The authors have made a comprehensive review about radiomics and differentiated thyorid cancer. The aim of the review is noble however the structure of the final paper and especially the conclusion drawn by the authors are in my view disapponting for the following reasons:

Point 1: The authors try to provide a very "all'-in-one" review about what is known for DTC but they spend almost half of the paper introducin generic aspects of radiomics not directly linked to DTC: in my view this is unnecessary

Response 1: Thank you for your comment. We have realized our introduction part is relative hard to read. We have added subheadings to divide introduction into 2 parts, which is more understandable and readable now. Some background introduction may be helpful to add knowledge and necessity to current review. 

Point 2: Reading the title of the paper the reader would expect to get information about the diagnosis of DTC but then the authors state they want to provide "insight into the capacity of radiomics to 24 predict lymph node (LN) metastasis, distant metastasis, tumor extrathyroidal extension, disease-25 free survival, and B-Raf proto-oncogene serine/threonine kinase (BRAF) mutation": in my view this is misleading

Response 2: Thank you for your suggestion. There are some exaggerated assumptions about this expression for a review. We have revised this sentence in abstract (red).

Point 3: The conclusion is totally absent becauese atuhors state that the "aforementioned findings"demonstrate the usefulness of radiomics...regardless of the image type": such sentence is lacking scientific evidence beacuse US is the most widely used technique and should receive in my view a special focus both in the discussion and in the conclusion.

Response 3: Thank you for your comment. We have re-organized the conclusion as both you and reviewer 5 suggested.

Reviewer 5 Report

Your review paper is well written with extensive review of published papers on radiomics related to thyroid cancer. Your responses to the  following comments are needed to improve your paper. 

Table 2 shows 74-86% diagnostic accuracy which raises a question as to whether a radiomics is cost-effective. Unless we can obviate FNAB which provides not only histological diagnosis but also gene profile we may not need a radiomics as US radiomics was negated in predicting BRAF mutation in DTC. Deep learning radiomics may be better with good and optima data but seems impractical for cost-effectiveness. Table 1 is about prediction which is not clear. Is prediction of recurrent tumor or prognosis of patients? How much improvement was mentioned with radiomics compared with no radiomics?

Author Response

Response to Reviewer 5 Comments

Your review paper is well written with extensive review of published papers on radiomics related to thyroid cancer. Your responses to the following comments are needed to improve your paper.

Point 1:

Table 2 shows 74-86% diagnostic accuracy which raises a question as to whether a radiomics is cost-effective. Unless we can obviate FNAB which provides not only histological diagnosis but also gene profile we may not need a radiomics as US radiomics was negated in predicting BRAF mutation in DTC. Deep learning radiomics may be better with good and optima data but seems impractical for cost-effectiveness. Table 1 is about prediction which is not clear. Is prediction of recurrent tumor or prognosis of patients? How much improvement was mentioned with radiomics compared with no radiomics?

Response 1: We have added this cost-effectiveness issue in both sections of limitations (line 444-446 in red) and conclusion (466-470 in red). And we have revised the Table 1 (in red). For improvement of radiomics relative to no radiomics, it various in different studies, but in general, the radiomics may more objective compared to radiologists experience. 

Round 2

Reviewer 4 Report

The authors have made an effort to improve the paper however my major concerns still remain.